# Association of MALAT1 and PVT1 Variants, Expression Profiles and Target miRNA-101 and miRNA-186 with Colorectal Cancer: Correlation with Epithelial-Mesenchymal Transition

**DOI:** 10.3390/ijms22116147

**Published:** 2021-06-07

**Authors:** Abdullah F. Radwan, Olfat G. Shaker, Noha A. El-Boghdady, Mahmoud A. Senousy

**Affiliations:** 1Department of Biochemistry, Faculty of Pharmacy, Egyptian Russian University, Cairo 11829, Egypt; abdullah-fathy@eru.edu.eg; 2Department of Medical Biochemistry and Molecular Biology, Faculty of Medicine, Cairo University, Cairo 12613, Egypt; olfatshaker@yahoo.com; 3Department of Biochemistry, Faculty of Pharmacy, Cairo University, Cairo 11562, Egypt; noha.elboghdady@pharma.cu.edu.eg

**Keywords:** adenomatous polyps, CRC, diagnosis, long non-coding RNA, microRNAs, SNPs

## Abstract

The influence of *PVT1* and *MALAT1* variants on colorectal cancer (CRC) susceptibility and their impact on *PVT1*/*miRNA-186*/epithelial-mesenchymal transition (EMT) and *MALAT1*/*miRNA-101*/EMT axes in CRC are unknown. We investigated the influence of *PVT1 rs13255292* and *MALAT1 rs3200401* on the risk of CRC and adenomatous polyps (AP), their impact on the long noncoding RNAs *PVT1* and *MALAT1* expression and their target *miRNA-186*, *miRNA-101*/E-cadherin pathways, along with their potential as early CRC biomarkers. Overall, 280 individuals were recruited: 140 patients with CRC, 40 patients with AP, and 100 healthy volunteers. Genotyping and serum expression profiles were assessed using qPCR. The EMT biomarker, E-cadherin, was measured by ELISA. *rs3200401* was associated with increased CRC risk, whereas *rs13255292* was protective. Serum *PVT1* and *MALAT1* were upregulated in CRC and AP patients versus healthy controls, whereas, *miRNA-186*, *miRNA-101* and E-cadherin were downregulated in CRC versus non-CRC groups. *MALAT1* showed superior diagnostic potential for CRC and predicted CRC risk among non-CRC groups in the multivariate logistic analysis. *PVT1*, *MALAT1*, *miRNA-186* and *miRNA-101* levels were correlated with E-cadherin, tumor stage, lymph node and distant metastasis. E-cadherin was lost in metastatic vs. non-metastatic CRC. *rs3200401CC* genotype carriers showed higher E-cadherin levels than *CC + CT* carriers. *rs3200401* was correlated with lymph node status. For the first time, *rs13255292* and *rs3200401* are potential genetic CRC predisposition markers, with *rs3200401* possibly impacting the EMT process. Serum *PVT1*, *MALAT1*, *miRNA-186* and *miRNA-101* are novel non-invasive diagnostic biomarkers that could improve the clinical outcome of CRC.

## 1. Introduction

Colorectal cancer (CRC) is among the most frequently diagnosed cancers worldwide, and it is a leading cause of cancer-related deaths around the world [1]. Advanced cases of CRC have poor prognosis and unsatisfactory survival rates [2]. We must make advances in the early detection and therapy of CRC to increase patient survival.

Because of the wide variety of biological processes that they play a part in, non-coding RNAs (ncRNAs) such as long ncRNAs (lncRNAs) and microRNAs (miRNAs) have recently received enough attention to merit mentioning. Mutations or the unnatural expression of ncRNAs are closely associated with many diseases, particularly cancer [3]. The crosstalk research between lncRNAs, miRNAs, and their master regulated proteins has become a newfangled passion for deciphering cancer’s molecular mechanism, including CRC.

An 8.5 kb lncRNA called lung adenocarcinoma transcript-1 (*MALAT1*) is located on chromosome 11q36. It has been reported that *MALAT1* significantly subsidizes CRC development, progression, metastasis and survival rate [4]. It also functions as a miRNA sponge in CRC [5]. For instance, it can foster epithelial-mesenchymal transition (EMT) progression, which in turn boosts tumor growth by acting as a competing endogenous RNA (ceRNA) for *miRNA-101* in CRC [6]. *miRNA-101* is one of the small ncRNAs that has been found to act as a tumor suppressor in different types of cancer by targeting oncogenes and anti-oncogenes [7]. Although an established inverse significant correlation between *MALAT1* and *miRNA-101* was stated in different forms of cancer [8,9], the clinical relevance of this correlation in CRC remains to be scrutinized.

One of the most enticing features of miRNAs is their capacity to act as therapeutic targets in different diseases, making them particularly effective in regulating different cell processes important to malignant cell homeostasis which have brought hope for cancer patients [7]. *miRNA-101* was found to target different pathways that promote breast cancer cell apoptosis by inhibiting the expression of Jak2, EYA1 and SOX2 acting as a potential therapeutic target in breast cancer [10]. In addition, *miRNA-186* acts as a therapeutic agent in human esophageal squamous cell carcinoma by targeting HOXA9 [11]. However, they still need trials in CRC.

Because of the worthiness of new lncRNAs that were discovered and the thorough investigation into their roles in different types of cancer, the recently discovered oncogenic factor, lncRNA plasmacytoma variant translocation 1 (*PVT1*), is one of the proven overexpressed factors in CRC [12,13] and other types of cancer [14,15]. Additionally, *PVT1* is related to miRNAs in cancer development. It is possible that *PVT1* acting as a sequester for *miRNA-186* leading to inhibition of its activities, affecting proliferation, invasion, and metastasis of cancer [15,16]. It is suggested that *miRNA-186* downregulates Twist1, leading to overexpression of E-cadherin, thus suppressing the EMT process [16,17]. Reduced expression of E-Cadherin can promote the EMT process, which leads to the development of a tumor [18]. There is a paucity of literature on the correlation between *PVT1* and *miRNA-186* as an important diagnostic and prognostic parameter in CRC and other cancer types. Uncovering this correlation would allow for a better understanding of the disease biology. It would serve as a useful indicator for predicting critical diagnostic and prognostic parameters in cases of CRC.

Single nucleotide polymorphisms (SNPs) in lncRNAs and miRNAs genes may affect the risk, prognosis and treatment response in CRC. Identifying miRNA:mRNA interactions might aid in understanding the functions of many unknown SNPs [19]. CRC risk factors identified through genome-wide association studies have also reported genetic variants in the genomic regions of lncRNAs. These SNPs altered the expression and/or structure of lncRNA, as well as affected the mechanisms of lncRNA [4,20]. *MALAT1 rs3200401* and *PVT1 rs13255292* SNPs were assessed with different types of cancer [21,22], but not yet extensively investigated in CRC.

Therefore, this study aimed to investigate the association of *MALAT1 rs3200401* and *PVT1 rs13255292* SNPs with the susceptibility of CRC. Furthermore, we explored the relationship of these polymorphisms with *MALAT1* and *PVT1* expression and their target *miRNA-101* and *miRNA-186*, respectively, in CRC. The impact of studied SNPs and the correlation of studied ncRNAs with E-cadherin as an EMT biomarker were also investigated. Moreover, we analyzed the correlations between studied parameters and the clinicopathological parameters of CRC and their potential in early diagnosis and prognosis of CRC.

## 2. Results

### 2.1. The Studied Groups’ Demographical and Clinicopathological Properties

As illustrated in Table 1, all of the various demographic, laboratory and pathological features of the examined groups are presented. Patients with AP are significantly younger than those with CRC (*p* < 0.0001) and healthy groups (*p* = 0.006). Gender was not significantly different (*p* = 0.86); however, a male preponderance in CRC and AP patients representing 64.2% and 62%, respectively, was observed. 30% and 20% of CRC and AP patients were tobacco smokers, respectively.

CRC cases included 11.4% poorly-differentiated, 70% moderately differentiated, and 18.6% well-differentiated tumors with regard to histopathological grading. Moreover, 73.5% of all CRC were located in the colon, while 26.5% were found in the rectum. Tumors varied in size (1.5 cm or greater). Overall, 93.5% of CRC tumors were adenocarcinoma. Only 17.1% of the patients had metastatic CRC, with hepatic focal lesions being present in all. Furthermore, 62.8% of CRC patients have been diagnosed with the American Joint Committee on Cancer (AJCC) early stages (I and II); however, 37.2% of the diagnosed CRC patients with late stages (III and IV). Approximately, 40% of the patients throughout the AP group had multiple (≥3) differential polyps. In contrast, the residual had either one or two polyps, and none of them were in one of the polyposis syndromes. In polyps, two-thirds were tubulovillous adenomas, with half showing dysplasia.

### 2.2. Association of rs3200401 (C/T) and rs13255292 (C/T) with the Risk of CRC and AP

Genotyping was handled without the participants being told that they were participating in a case-control analysis. For *rs3200401* and *rs13255292* SNPs, MAF in the controls was *T* = 0.34 and *T* = 0.42, respectively, which were slightly higher than the global MAF (*T* = 0.14 for *rs3200401* and *T* = 0.20 for *rs13255292*), but was still close to the highest population MAF for both SNPs *rs3200401 T* = 0.31 and *rs13255292 T* = 0.43, reported in Ensembl release 102-November 2020 (Appendix A). The distribution of the *rs3200401* and *rs13255292* genotypes in control and patient groups did not stray significantly from HWE (*p* > 0.05) (Appendix A).

The allele frequencies and genotypes for *rs3200401* and *rs13255292* are displayed in Table 2. For *rs3200401*, the minor *T* allele was a 2.43-fold candidate risk factor for CRC (Tvs. *C*, adjusted OR = 2.43, *p* < 0.0001), as revealed in the allelic model. The genotype and allele frequencies of the major *C* and minor *T* alleles were not significantly different between the AP patients and controls (*p* > 0.05) (Appendix A). In CRC patients, the genotype and allele frequencies for *rs3200401* (*C/T*) in the codominant model was significantly different than in healthy controls (*CC*, *CT*, *TT*: 22.1%, 44.3%, 33.6% in CRC patients vs. 42%, 48%, 10% in controls) with a minor homozygote *TT* genotype that showed a 6.79-fold increased risk of CRC (adjusted OR = 6.79, *p* < 0.0001). Moreover, the *CT* + *TT* (dominant model) and *TT* (recessive model) genotypes showed a 2.62 and 4.82-fold increased risk of CRC (adjusted OR = 2.62 and 4.82, respectively, *p* < 0.0001) after adjustments for age and sex (Table 2).

Regarding *rs13255292*, the genotype and allele frequencies were not significantly different between the AP patients and controls (*p* > 0.05) (Appendix A). However, the minor *T* allele was a candidate protective factor against the risk of CRC by 0.66-fold (*T* vs. *C*, adjusted OR = 0.66, *p* = 0.04), as revealed in the allelic model. In CRC patients, the genotype frequencies for *rs13255292* in the codominant model were significantly different than in healthy controls (*CC*, *CT*, *TT*: 45.7%, 44.3%, 10% in CRC patients vs. 31%, 55%, 14% in controls) with a heterozygote *CT* genotype that showed a 0.56-fold decreased risk of CRC (adjusted OR = 0.56, *p* = 0.086). Furthermore, the *CT* + *TT* (dominant model) genotypes showed a 0.54-fold decreased risk of CRC (adjusted OR=0.54, 95% CI = 0.32–0.94, *p* = 0.021) after adjustments for age and sex (Table 2).

### 2.3. Association of MALAT1 rs3200401 and PVT1 rs13255292 with CRC vs. Non-CRC

The evaluation of the genotype and allele frequencies between CRC and non-CRC groups showed the same models’ significance for *MALAT1 rs3200401* against the genotype and allele frequencies evaluation between CRC vs. healthy control groups. However, the genotype and allele frequencies during the evaluation of *PVT1 rs13255292* in CRC vs. non-CRC groups showed only a significant difference in the dominant model (*CT* + *TT* vs. *CC*) as a protective predictor adjusted OR = 0.59, *p* = 0.034 (Table 2 and Table 3).

### 2.4. Selection of the Best Fit Models

Non-nested models can be compared using Akaike’s Information Criteria (AIC) and Bayesian Information Criteria (BIC) calculations. A lower AIC and BIC means that the model is more likely to be close to the model that fits the data best or the model that is most likely to predict results. For *rs3200401*, the allelic model represented the best fit model when comparing CRC vs. healthy control groups and CRC vs. non-CRC. For *rs13255292,* the dominant model was the best fit model when comparing CRC vs. healthy control and CRC vs. non-CRC (Table 2 and Table 3).

### 2.5. Haplotype Analysis

We looked at the combined effect of the analyzed gene polymorphisms regarding CRC risk (Table 4). We found that the *rs13255292-rs3200401 CT* haplotype was associated with increased CRC risk by 2.21-fold (*CT* vs. *CC* haplotype, adjusted OR = 2.21, *p* = 0.0032). Other haplotypes were not statistically associated with CRC risk (*TC* vs. *CC*, adjusted OR = 0.64, *p* = 0.17, *TT* vs. *CC*, adjusted OR = 1.5, *p* = 0.13).

### 2.6. Serum Expression Levels of MALAT1, PVT1, miRNA-101 and miRNA-186 in CRC and AP

Serum *MALAT1* was significantly upregulated with a median (IQR) fold change of 102.5 (35.24–136.8) (*p* < 0.0001) and 20.50 (11.44–39.87) (*p* = 0.0234) in CRC and AP patients, respectively, compared to healthy controls. Moreover, serum MALAT1 levels were significantly higher in CRC than AP patients (*p* = 0.0232) (Figure 1A). Furthermore, serum *PVT1* expression was upregulated significantly in CRC and AP patients compared to healthy controls, with a median (IQR) fold change of 117.6 (29.96–279.8) (*p* < 0.0001) and 23.03 (6.75–49.29) (*p* = 0.0148), respectively. Besides, serum *PVT1* amounts were elevated significantly in CRC patients than AP (*p* = 0.0391) (Figure 1B).

Serum *miRNA-101* and *miRNA-186* were downregulated significantly in CRC patients compared to healthy controls, with a median (IQR) fold change of 0.263 (0.0074–0.8587) (*p* < 0.0031) and 0.1398 (0.01044–0.7203) (*p* = 0.0236), respectively (Figure 1C,D). Serum *miRNA-101* and *miRNA-186* levels were quantitatively lower in CRC than AP patients but did not reach statistical significance (*p* > 0.9) and (*p* = 0.1577), respectively. By joining the AP group with the control group and comparing them with the CRC group, serum *miRNA-101* and *miRNA-186* were significantly lower in CRC compared to non-CRC groups (AP + healthy controls) (*p* = 0.0045) and (*p* = 0.0025) (Figure 1E,F).

### 2.7. Serum Levels of E-Cadherin

E-cadherin revealed, in CRC, a significant difference from healthy control group (*p* < 0.0001), as illustrated in Figure 2A. However, E-cadherin failed to reveal a significant difference between the AP group and both the CRC group (*p* = 0.1095) and healthy control groups (*p* < 0.1842). Interestingly, it showed a more substantial decrease in CRC than in non-CRC groups (*p* < 0.0001) (Figure 2B).

When it comes to comparing E-cadherin levels within the CRC group between the metastasized and non-metastasized patients, we found a loss in E-cadherin expression in the CRC-metastatic group than the CRC-nonmetastatic one (*p* = 0.0049).

### 2.8. Association of rs3200401 and rs13255292 with Serum MALAT1, PVT1, miRNA-101, miRNA-186 and E-Cadherin Levels in CRC Patients

To investigate the mechanistic role of *rs3200401* and *rs13255292* in CRC, we determined serum *MALAT1*, *PVT1*, *miRNA-101*, *miRNA-186* and E-cadherin levels in CRC patients carrying different SNP genotypes (Figure 3). We found that serum *MALAT1* expression level was higher in the *TT* genotype carriers of *rs3200401* than in the *CC* as well as *TT* + *CT* genotype carriers than in *CC* genotypes carriers but without reaching the statistical significance (*p* > 0.05) (Figure 3A), while serum *PVT1* expression level was lower in the *TT* genotype carriers than in the *CC* and higher in *CC* genotype than *CC* + *CT* genotypes carriers, but without reaching statistical significance (*p* > 0.05) (Figure 3B). Regarding *miRNA-101*, we failed to find a significant difference in its expression among CRC patients with different *rs3200401* genotypes (*p* > 0.05) (Figure 3C). Additionally, *miRNA-186* expression failed to have a significant difference in its expression levels in the CRC group with different *rs13255292* genotypes; however, *TT* genotype carriers were higher than *CC* and *CC* + *CT* genotypes (*p* > 0.05) (Figure 3D).

Referring to E-cadherin, there was a higher significant difference in E-cadherin expression levels in *CC* genotype carriers *rs3200401* genotypes than *CC* + *CT* genotypes carriers (*p* = 0.039) (Figure 3F), suggesting an effect of *rs3200401* on the EMT process. However, we failed to find a significant difference in its expression among CRC patients with different *rs13255292* genotypes (*p* > 0.05) (Figure 3E).

### 2.9. Correlation between rs3200401 and rs13255292 and the Clinicopathological Characteristics

Association analyses between *MALAT1 rs3200401* and anatomical site, TNM stage, lymph node status and metastasis were performed. *MALAT1 rs3200401* showed significant correlation with the lymph node status only for the current models (codominant, recessive and log additive model). However, *PVT1 rs13255292* was not significantly associated with the clinicopathological parameters of CRC (adjusted OR > 0.05) (Appendix A).

### 2.10. Diagnostic Performance of the Studied Parameters between the Studied Groups

Regarding lncRNAs, ROC analysis revealed that serum *MALAT1* and *PVT1* distinguished patients with CRC from healthy controls with an AUC = 0.965, 95% CI = 0.9261 to 0.991, *p* < 0.0001, with sensitivity of 89%, specificity of 95% at a cutoff > 10.70-fold and an AUC = 0.915, 95% CI = 0.8441 to 0.9859, *p* < 0.0001, with sensitivity of 90%, specificity of 95% at a cutoff > 13.96-fold, respectively.

Serum *MALAT1* and *PVT1* also discriminated CRC patients from AP with an AUC = 0.823, 95% CI = 0.7210 to 0.9255, *p* = 0.0004, with sensitivity of 74%, specificity of 92% at a cutoff > 55.6-fold and AUC = 0.769, 95% CI = 0.6581 to 0.8795, *p* = 0.0010, with sensitivity of 60%, specificity of 94% at a cutoff > 73.9-fold, respectively.

Furthermore, serum *MALAT1* and *PVT1* discriminated AP from healthy controls with an AUC = 0.965, 95% CI = 0.7639 to 0.9974, *p* < 0.0001, with sensitivity of 76%, specificity of 95% at a cutoff > 10.9-fold and an AUC = 0.935, 95% CI = 0 0.6581 to 0.8795, *p* < 0.0001, with sensitivity of 60%, specificity of 94% at a cutoff > 73.9-fold, respectively (Figure 4A–F).

Referring to miRNAs, ROC analysis showed that they discriminated CRC from healthy controls with an AUC = 0.747, 95% CI = 0.6238 to 0.8693, *p* = 0.0013, with sensitivity of 62%, specificity of 76% at a cutoff < 0.2888-fold for *miRNA-101* and an AUC = 0.698, 95% CI = 0.5705 to 0.8247, *p* = 0.0087, with sensitivity of 54%, specificity of 77% at a cutoff < 0.204-fold for miRNA-186 (Figure 4G,H).

Denoting to E-cadherin, ROC analysis revealed that it could discriminate CRC from healthy controls with an AUC = 0.892, 95% CI = 0.8189 to 0.9646, *p* < 0.0001, with sensitivity of 70%, and specificity of 95% at a cutoff < 3.661 ng/mL (Figure 4I).

### 2.11. Prognostic Significance of E-Cadherin in CRC

E-cadherin ROC analysis discriminated metastatic from non-metastatic patients within CRC group with AUC = 0.835, 95% CI = 0.5721 to 0.8504, *p* = 0.0007. The sensitivity was 81.8% and thespecificity was72.7% at a cutoff < 2.86 ng/mL (Figure 5).

### 2.12. Diagnostic Performance of the Studied Parameters between CRC and Non-CRC

By comparing AUCs between CRC vs. non-CRC, it could be seen that *MALAT1*, *PVT1* and E-cadherin with AUC = 0.907, 0.848 and 0.864, respectively, represented diagnostic performance superior to *miRNA-101* and *miRNA-186* with AUC = 0.686 and 0.702, respectively, in CRC diagnosis (Table 5).

### 2.13. Logistic Regression Analysis of the Studied Parameters

Univariate and multivariate logistic regression analyses were performed to select the predictor parameters associated with CRC risk among non-CRC groups diagnosis (Table 6). Expression levels of *MALAT1*, *PVT1*, *miRNA-101*, *miRNA-186* and E-cadherin were selected as significant predictors associated with the chances of CRC diagnosis in the univariate analysis (*p* < 0.05). In a stepwise forward multivariate analysis, only *MALAT1* turned out to be a significant predictor of the risk of being diagnosed with CRC (*p* = 0.0064).

### 2.14. Correlation between the Studied Parameters and the Clinicopathological Characteristics

In CRC group, we found a significant inverse correlation when it comes to *MALAT1* vs. *miRNA-101* (r = −0.4025, *p* = 0.006) and *PVT1* vs. *miRNA-186* (r = −0.4688, *p* = 0.002). A significant positive correlation was found between *MALAT1* and *PVT1* in the CRC group (r = 0.3608, *p* = 0.014). We also found inverse correlations between *MALAT1* and *PVT1* with E-cadherin (r = −0.3236, *p* = 0.0343), (r = −0.3078, *p* = 0.0447), respectively, and positive correlations between *miRNA-101* and *miRNA-186* with E-cadherin (r = 0.3559, *p* = 0.0207), (r = −0.4688, *p* = 0.001), respectively (Figure 6) (Appendix A).

In addition, there was a positive correlation between *MALAT1* and the clinicopathological parameters in CRC patients e.g., tumor stage (r = 0.3340, *p* = 0.0231), lymph node status (r = 0.3600, *p* = 0.019) and metastasis (r = 0.3062 *p* = 0.0385), respectively. A positive correlation between *PVT1* with the tumor stage (r = 0.4412, *p* = 0.0021), the lymph node status (r = 0.3181, *p* = 0.0312) and the distant metastasis (r = 0.3018, *p* = 0.0415) of the CRC patients, respectively. Furthermore, a significant inverse correlation was found between *miRNA-101* and *miRNA-186* against stage (r = −0.4132, *p* = 0.0048), (r = −0.2976, *p* = 0.0588); lymph node status (r = −0.3390, *p* = 0.0227), (r = −0.1790, *p* = 0.2627); metastasis (r = −0.3377, *p* = 0.0233), (r = −0.4011, *p* = 0.0094), respectively (Figure 6) (Appendix A).

## 3. Discussion

There is an urgent need to investigate new biomarkers that could work as a robust panel for CRC diagnosis and screening. Carcinoembryonic antigen (CEA), the current conventional tumor marker in CRC management, is most elevated in the late metastatic stages. It has low sensitivity and specificity when used to distinguish early non-metastatic stages [23]; thus, it is recommended to be used as a biomarker for already confirmed metastasized diagnosed CRC. In clinical practice, the measurement of CEA proved to be a well-established biomarker, most useful in determining distant metastases [24] and monitoring the metastatic disease’s response to systemic therapy and detecting the recurrence of CRC [25]. Therefore, it is primarily used for prognosis [26] and in the surgical planning of CRC, not for new and early diagnosis [27]. Therefore, new diagnostic and prognostic markers are needed.

There is a lack of studies examining modifications in the expression of regulatory lncRNAs and their possible interaction with miRNAs. It is essential to clarify some miRNAs critical roles and their correlated potential host lncRNAs and explore colon cancer regulatory networks.

To date, the study of the SNPs role of *MALAT1 rs3200401* and *PVT1 rs13255292* in various oncological processes development is still unclear. This is particularly important when it comes to investigating their evidence on serum *MALAT1* and *PVT1*; and their crosstalk with their inversely correlated miRNAs, *miRNA-101* and *miRNA-186*. Besides their possible indirect effect on the EMT process in CRC patients.

The current study has demonstrated that genetic variants that include *MALAT1 rs3200401* and *PVT1 rs13255292* exhibited effects on the development and predisposition to CRC, but not the formation of the adenomatous polyp. Interestingly, the haplotype containing the two risk alleles of both SNPs was associated with increased CRC risk. In addition, serum *MALAT1*, *PVT1*, *miRNA-101* and *miRNA-186* expression levels seemed to be associated with these SNPs, but did not reach the significance level. These findings may contribute to the diverse nature and pathology of CRC and include these SNPs as possible genetic susceptibility markers for sporadic and non-inherited CRC via functional intonation of the expression of lncRNAs.

Firstly, to the best of our knowledge, the findings have shown that *T* allele carriers in *PVT1 rs13255292* confer a protective effect against the growth and progression of CRC. Only one study testified *rs13255292* in diffuse large B-cell lymphoma (DLBCL); by contrast, it was found that *T* allele *PVT1 rs13255292* was the risk of DLBCL [28]. We need more studies to elaborate the exact *rs13255292* action in different populations.

As far as we know, this is the first research to assess the *MALAT1 rs3200401* variant in CRC. Patients in our research carrying the *T* allele in the *MALAT1 rs3200401* variant were found to associate with the risk of CRC. The finding is harmonized with the studies that indicated that *T* allele carriers have a risk of different malignancies such as prostate cancer [29], esophageal squamous cell carcinoma [30] and gastric cancer [31]. However, other studies have shown that patients who are major T allele carriers can explain the decreased aggressiveness toward tumors, while also explaining the improved survival rates in various cancer forms, such as breast cancer [32] and advanced lung adenocarcinoma patients [33]. These discrepant results might be because of the assessment of other kinds of cancer and diverse populations.

The lncRNA SNP database, as stated by a research study, was used to envisage the possible roles of *rs3200401* [34]. The *C* and *T* variation of *rs3200401* events may modify the structural peculiarities of *MALAT1*, leading to a weakened interaction between *MALAT1* and its binding protein SRSF2 (serine and arginine rich splicing factor 2) as a protein-coding gene [6]. Besides, *MALAT1* was testified to accompany phosphorylation of SRSF2, interaction with SR proteins (i.e., RNA-binding proteins) as a “molecular sponge,” and alternate splicing pre-mRNAs regulation. Altogether, it was biologically conceivable that SNP *rs3200401 C* and *T* alleles could remodel cancer-associated genes’ expression degrees, consequently participating in the carcinogenesis and progression of cancer [30,35]. However, *rs13255292* is identified in limited numbers of literature; consequently, its functional role and molecular mechanism regarding the alteration of *PVT1* features remain unclear.

*MALAT1* and *PVT1* were recognized as oncogenic lncRNAs in various malignancies, they have been associated with reduced survival duration in several studies [33,36]. Thus, their biological roles have recently attracted interest in the production and advancement of CRC. In CRC patients, expression levels of *MALAT1* and *PVT1* were overexpressed compared to AP and healthy control, indicating a poor prognosis. However, only an association between *MALAT1 rs3200401* and the lymph node status, suggesting that *rs3200401* has a prognostic value. However, *PVT1 rs13255292* and the clinicopathological factors were not related.

Nevertheless, there are rare reports of *MALAT1* and *PVT1* pathways involved in the tumorigenesis and growth of CRC. As a critical cellular program characterized by the loss of epithelial characteristics and mesenchymal phenotype acquisition, EMT is always identified as the crucial step of metastasis. Nonetheless, *MALAT1* and *PVT1* were reported through their sponging function, and have a critical regulating role on EMT [35,37], thereby promoting metastasis and CRC progression.

This study revealed that *MALAT1* and *PVT1* were overexpressed in CRC and negatively correlating with E-cadherin as one of the EMT process indicative markers and part of an EMT regulatory network, indicating that *MALAT1* and *PVT1* could act as potential prognostic markers. It has been ascertained that the phase of EMT is correlated with poor results and is valuable as a CRC prognostic indicator [38].

We found that *miRNA-101* with E-cadherin decreased significantly in the CRC group more than AP and healthy control, which was matched with preceding research which reported that *MALAT1* induces EMT through different mechanisms, e.g., the Wnt/β-catenin [35], Ezh2-Notch1 [39] and TGF-β signaling pathways [40]. Furthermore, it was identified that *miRNA-101* could bind to complementary sequences in *MALAT1* [41]. Increased expression of *miRNA-101* contributes to the downregulation of *MALAT1*; upregulation of *MALAT1* acting as an endogenous sponge gene decreased the expression of *miRNA-101* in glioma [41] and liver fibrosis [42]. Because TGF-β was found to be a potent EMT inducer, CRC cells treated with TGF-β reported microscopic morphological changes consistent with EMT and showed decreased levels of E-cadherin [43]. Furthermore, because *miRNA-101* plays a crucial role in TGF-β modulation [41,44], we assume that the upregulation of *MALAT1* leads to the subsequent downregulation of *miRNA-101*, promoting the expression of TGF-β, leading to eventually diminishing E-cadherin, promoting metastasis and CRC progression.

In our research, *miRNA-186* with E-cadherin concomitantly downregulated significantly in the CRC group more than AP and healthy control, indicating the promotion of CRC invasion, migration and metastasis. These findings are agreed with the reported assay of *PVT1* functions in EMT, cancer metastasis and migration [16,45]. It has been established that *PVT1* contributed to the *PVT1*/*miRNA-186*/Twist1 regulatory, confirming that *PVT1* endorses the expression of Twist1 via its knockdown role, which is a transcription factor linked to the EMT process, thus promoting the EMT [37].

Regarding the studied markers’ diagnostic performance, circulating lncRNAs and miRNAs are easily accessible, valid and accurate genetic tests in different types of cancer, including CRC [46,47,48]. Here, we observed that serum *MALAT1*, *PVT1*, *miRNA-101* and *miRNA-186* were distinctively expressed between individuals with CRC and healthy control and/or non-CRC, besides being distinguished against CRC from other groups with moderate to high sensitivity and specificity, indicating serum *MALAT1*, *PVT1*, *miRNA-101* and *miRNA-186* as potential novel biomarkers for early and new CRC diagnosis. Nevertheless, serum *MALAT1*, *PVT1*, *miRNA-101* and *miRNA-186* in the CRC against non-CRC groups were significantly upregulated, and ROC analysis differentiated the two groups. Remarkably, *MALAT1* and *PVT1* showed higher accuracy, superior sensitivity and specificity than *miRNA-101* and *miRNA-186*. These findings associate serum *MALAT1* and *PVT1* as reliable non-invasive early biomarkers and promising therapeutic targets for CRC treatment. Although the combination of *MALAT1* and *PVT1* with other tumor markers may improve the early CRC diagnosis, this needs further investigation.

Even so, a significant positive association was also observed between serum *MALAT1* and *PVT1*, indicating their concomitant expression in CRC. Besides, associations between these lncRNAs and CRC clinicopathological parameters were found, such as stage, nodal and distant metastases, suggesting that *MALAT1* and *PVT1* play a crucial role in directly contributing to tumor progression. Consistent with other similar studies that reported *MALAT1* and *PVT1* were correlated with CRC nodal and/or distant metastases and different types of cancer [36,49,50]. The identification of *MALAT1* and *PVT1* downstream targets played an essential role in assuming their probable mechanisms. Previous studies have stated that *MALAT1* targets RUNX2, β-catenin, AKAP-9 and Akt/mTOR signaling [51,52,53] and *PVT1* targets MYC, YAP1 and LASP1 [54,55], respectively, promoting tumorigenesis.

Our research maintained a substantial negative association between serum *miRNA-101* and *miRNA-186*, CRC tumor-related records, stage, nodal, and metastases. These results confirmed that *miRNA-101* and *miRNA-186* were tumor suppressive miRNAs in CRC. What is agreed with different studies that reported *miRNA-101* and *miRNA-186* were correlated inversely with stage, nodal and/or CRC distant metastases and other tumors [10,11,56]. As miRNAs typically use their influence on their downstream targets, the identification of *miRNA-101* and *miRNA-186* downstream targets played an essential role in understanding their probable mechanisms. The *miRNA-101* has been documented to target EZH2, c-FOS, CXCR7, Rac1, COX2 and SOX9 and *miRNA-186* has been reported to target YAP1, NR5A2, MTSS1 and NSBP1 in different kinds of malignancies [10,57], thus, suppressing cell proliferation, inducing apoptosis, thus acting as promising therapeutic targets.

Referring to the diagnostic and prognostic performance of E-cadherin, it exhibited high sensitivity and specificity, showing potential diagnostic and prognostic performance. Moreover, it showed a positive correlation with *miRNA-101* and *miRNA-186* and an inverse correlation with *MALAT1* and *PVT1*; these correlations may prove that a decrease in E-cadherin expression in the primary tumor is correlated with the ability of the tumor to spread, implying *PVT1*/*miRNA-186* and *MALAT1*/ *miRNA-101* in this context. This may be a prognostic factor for the further development of cancer.

However, the gold standard for CRC screening is colonoscopy; new and efficient non-invasive diagnostic and prognostic biomarkers are better and simpler to use than the invasive method of a colonoscopy, which appears to be reproducible and cost-effective. Noticeably, the limitations of our study should not be ignored. The sample size of our research which includes 280 volunteers may led to a limited statistics power. Thus, this research needed to be testified on a broader scale or population with a larger number of participants and different racial groups.

In conclusion, this study is the first to assess *MALAT1 rs3200401* and *PVT1 rs13255292* SNPs in CRC and introduce them as genetic biomarkers of CRC. Furthermore, lncRNAs *MALAT1* and *PVT1* may act as robust diagnostic and prognostic markers; furthermore, *miRNA-101* and *miRNA-186* may serve as markers with powerful diagnostic and predictive capabilities. Notwithstanding, our results involve *MALAT1 rs3200401* and *PVT1 rs13255292* as potential genetic markers of CRC predisposition. *MALAT1* is an independent predictor and could be of clinical value in CRC diagnosis.

Among non-CRC classes, *MALAT1 rs3200401* could predict the risk of CRC diagnosis. CRC screening, genetic therapy, and hope for large-scale use are potential implications of our data. Eventually, the association of the studied ncRNAs with CRC environmental risk factors should be assessed in the future.

## 4. Materials and Methods

### 4.1. Patients

This hospital-based case control research paper included 280 individuals classified as 140 CRC cases, most of them with adenocarcinoma type, 40 patients with adenomatous polyps (AP), and 100 cancer-free controls. A follow-up colonoscopy and the positive results corroborated by pathology confirmed that the recruited personnel were assorted. All participators who did attend the Gastrointestinal (GI) Endoscopy Unit in Kasr Al-Ainy Hospital, Cairo University were all grownup (>18 years old), CRC and AP groups (*n* = 180). Colonoscopy was recommended for screening symptoms of the lower GI tract, including chronic constipation and diarrhea, alternating or mixed-type irritable bowel syndromes and rectum bleeding, which may indicate the presence of CRC. Additionally, worry of CRC symptoms and signs, such as severe unexplained weight loss and unexplained anemia, were addressed.

Each patient’s clinical data were fully compiled, including their full medical history, blood count, sedimentation rate of erythrocyte (ESR), fecal occult blood test, and biochemical liver profile. CRC patients’ clinical characteristics, including TNM stage, lymph node status, metastasis level, and tumor grade, were also reviewed from health records documentation. In addition, the differentiation criteria between metastatic and non-metastatic patients depend on computerized tomography (CT) scan, positron emission tomography (PET) scan or magnetic resonance imaging (MRI). In-person interviews were conducted to collect epidemiological information among first-degree relatives, such as gender, age, smoking status, and cancer history. The primary clinical data for all participants are included in Table 1. Patients who had cancer at any other site, previously undergoing chemotherapy and/or radiotherapy for CRC, and/or having been diagnosed with inflammatory bowel disease (IBD) were rejected.

Compatible controls were of matched age and sex for the patient population. Malignancy, polyps, or IBD were screened during the colonoscopy, and the test results of healthy controls were therefore negative. They had no genetic predisposition to hereditary adenomatous polyposis and familial non-polyposis CRC. Simultaneously, as CRC patients were registered, they were randomly chosen from physical examination populations in the same hospital.

The whole study and experimentations have been done in compliance with the applicable regulations and guidelines. All patients and controls or their legal representatives received a formal informed consent agreement. The Ethics Committee of the Faculty of Pharmacy, Cairo University approved the study protocol under approval no. (BC2553) and the informed consent in correspondence to the ethical principles of the Declaration of Helsinki.

### 4.2. Blood Collection and Storage

Blood samples of sex milliliters were withdrawn and separated into two vacutainers. The first 3 mL were collected into EDTA vacutainers for DNA extraction and genotyping; the rest of the blood was preserved in yellow gel vacutainers; after 30 min, the yellow vacutainers were subjected to centrifugation at 4000 rpm for 10 min to get the sera separated from clotted whole blood. Sera were divided into two aliquots, the first aliquoted sera were utilized for RNA extraction, and the other ones were used for the assay of E-cadherin. All aliquots were kept frozen at −80 °C until use.

### 4.3. SNPs Position

The substitution of cytosine with thymine at the 65504361 site of chromosome 11 is the essence of the *rs3200401* SNP of the *MALAT1* gene, whereas the substitution of cytosine with thymine at the 128064327 site of chromosome 8 is the essence of the *rs13255292* SNP of the *PVT1* gene. In addition, *rs3200401* is considered a noncoding transcript variant in contrast to *rs13255292*, which is considered an intron variant.

### 4.4. Selection of SNPs

We elected SNPs depending on the criteria of the global minor allele frequency MAF > 0.1, reported functional resemblance with their product, and previously reported correlations with cancer, knowing that the selected SNPs were not extensively studied in CRC and its related hallmarks. According to the SNPedia and NCBI dbSNP database, there are 24 prevalent SNPs located on the *MALAT1* gene. Almost all of them have the MAF < 0.10 according to Ensembl release 102—November 2020. However, *rs3200401*, *rs4102217*, *rs591291*, *rs1194338*, *rs7133268* and *rs7763881* are the only 6 SNPs that have global MAF > 0.1; of which the *rs3200401* was the only relevant SNP. Furthermore, the lncRNA *MALAT1* has many reported functional analyses with *rs3200401* in many cancer types, such as gastric cancer, bladder cancer, lung carcinoma and breast cancer. Furthermore, *rs3200401* had an apparent influence on the secondary structural characteristics and stability of *MALAT1* [58] and relevant to CRC [32]. Referring to *PVT1* SNPs, *rs13255292*, *rs2608053* and *rs1561927* are the only SNPs with MAF > 0.1 *rs13255292* is the most functional reported one, and it is reported in a closely related disease, which is ovarian cancer [22].

### 4.5. DNA Extraction and Genotyping

From whole EDTA blood samples, genomic DNA was extracted from all participants using the QIAamp DNA Mini Kit, according to the manufacturer’s instructions (Qiagen, Valencia, CA, USA). The yield purity and quantity were measured by the NanoDrop 2000 c model (Thermo Fisher Scientific, Waltham, MA, USA). Utilizing pre-designed primer/probe sets for *rs3200401* (*C*/*T*) [Assay ID: C_3246069_10, Catalog number: 4351379] and *rs13255292* (*C*/*T*) [Assay ID: C_3023274_10, Catalog number: 4351379] (Thermo Fisher Scientific, Waltham, MA, USA), genotyping was conducted using real-time PCR with the TaqMan allelic discrimination assay. DNA amplification was performed using a TaqMan Master Mix, containing 12.5 μL of TaqMan and 1.25 μL of TaqMan and probe/primer solution in a volume of 25 μL, one μL of DNA, and 10.25 μL of H_2_O. Real-time PCR was conducted applying Qiagen Rotor-Gene Q System; the following was implemented: 95 °C for ten minutes, then cycles of 92 °C for fifteen seconds and 60 °C for ninety seconds.

### 4.6. Assay of Serum MALAT1, PVT1, miRNA-186 and miRNA-101 by RT-qPCR

The miRNeasy extraction kit (Qiagen, Valencia, CA, USA) was used to deduce the total RNA from 200 μL hemolysis-free serum using one mL of QIAzol lysis reagent, as directed by the production company. To assess RNA concentration and purity, the NanoDrop 2000 c model (Thermo Fisher Scientific, Waltham, MA, USA) was used. In both lncRNA and miRNA expression analysis, the extracted RNA was used.

In a final volume of twenty μL RT reactions using the RT^2^ first strand kit (Qiagen, Valencia, CA, USA), the reverse transcription (RT) for *MALAT1* and *PVT1* was performed using 0.1 μg of total RNA, as instructed by the manufacturer. The RT products were diluted with fifty µL RNase-free water before real-time PCR. Serum expression of the studied lncRNAs was assessed employing GAPDH as an internal control using custom-made primers and the PCR Maxima SYBR Green kit (Thermo Fisher Scientific, Waltham, MA, USA) as previously described [59]. The primers used in this study were as follows: *MALAT1* forward, 5′-GCAGGGAGAATTGCGTCATT-3′ and reverse, 5′-TTCTTCGCCTTCCCGTACTT-3′ [35]; *PVT1* forward, 5′- TGAGAACTGTCCTTACGTGACC-3′, reverse, 5′-AGAGC ACCAAGACTGGCTCT −3′ (Invitrogen) [60]. Moreover, the primer sequences of GAPDH were forward, 5′-CCCTTCATTGACCTCAACTA-3′, reverse, 5′-TGGAAGATGGTGATGGGATT-3′. In short, real-time PCR was performed on twenty μL of reaction mixtures prepared by mixing ten μL of the master mix, one μL of the forward primer, one μL of reverse primer, 2.5 μL of dilute cDNA and 5.5 μL of RNase-free water with the Qiagen Rotor-Gene Q System. The program was run for ten minutes at 95 °C, followed by 45 cycles for fifteen seconds at 95 °C and sixty seconds at 60 °C. Cycle threshold (Ct) is the number of cycles needed to reach the fluorescent signal above real-time PCR threshold. For relative quantification, the fold change was calculated using 2^−∆∆Ct^.

The miScript II RT Kit (Qiagen) was used to perform RT for *miRNA-186* and *miRNA-101*. The cDNA samples were amplified according to the manufacturer’s instructions using the miScript SYBR Green PCR kit (Qiagen), the supplied miScript Universal Primer (reverse primer), along with *miRNA-186* and *miRNA-101* specific primers (forward primer). Shortly, real-time PCR was performed in twenty μL reaction mixtures where 5.5 μL of RNase free water, ten μL of miScript SYBR Green PCR Master Mix, and two μL of miScript forward and reverse primers were mixed with 2.5 μL of appropriately diluted cDNA template. In the Rotor-Gene Q equipment, PCR runs were performed under the following conditions: 95 °C for thirty minutes, followed by 40 cycles at 94 °C for fifteen seconds, 55 °C for thirty seconds, and 70 °C for thirty seconds. As previously mentioned, miRNA *SNORD68*, the housekeeping miScript PCR control, was used as an endogenous control [20].

### 4.7. Assessment of Serum E-Cadherin by Enzyme-Linked Immunosorbent Assay (ELISA)

A human E-cadherin ELISA kit (ab233611) for the quantitative assessment of human E-cadherin was provided by Abcam (Trumpington, Cambridge, UK).

### 4.8. Statistical Methods

Data were analyzed using GraphPad Prism 9.0 statistics software (San Diego, CA, USA) and SPSS software (Chicago, IL, USA) version-25. Qualitative data were expressed as number and percentage. while numerical data were described in terms of mean ± SD and median, interquartile range (IQR) or range as appropriate. Testing for normality was applied using Kolmogorov–Smirnov, D’Agostino and Pearson, and Shapiro–Wilk tests. We compared normally distributed variables using student’s *t*-test or one-way ANOVA followed by Tukey’s post hoc test when appropriate. However, not normally distributed variables, their comparisons tested using the Mann–Whitney U test or the Kruskal–Wallis test followed by a post hoc test if significant when appropriate. For genotypes investigation in controls and patient groups, the Hardy–Weinberg equilibrium (HWE) was used, complying with a goodness-of-fit chi-square test. When appropriate, Chi-square or Fisher’s exact tests were used to compare categorical data. Associations between SNPs and CRC risk have been evaluated in unconditional logistic regression models by odds ratios (ORs) and corresponding 95% confidence intervals (95% CIs). Spearman’s rho correlation coefficient were calculated as appropriate to evaluate the relationship between the measurements. Receiver-operating-characteristic (ROC) curve was used to evaluate the studied parameters’ diagnostic accuracy, and the area under the curve (AUC) was determined. When AUC < 0.6, it was regarded as inconsequential; on the other hand, between 0.7–0.89, a potential or promising discriminator was considered, even though AUC > 0.9 was supposed to be a significant discriminator. SNPs analysis, SNPs correlation with clinicopathological parameters and haplotype analysis (allelic combinations) were carried out using the online SNPstats software (https://snpstats.net/, accessed on 15 May 2021) in logistic regression models. The variables of age and sex were used to account for variation in SNP results due to outside influences. For the adjustment of the data to the confounding factors, age and sex were included as covariates. Significant predictor variables in the univariate analysis were included in a stepwise forward multivariate analysis to determine the final predictor variables for the probability of being diagnosed with CRC. An internal 10-fold cross-validation was conducted to confirm the reproducibility of the results. When the *p*-value is equal to or less than 0.05, we regard the results as statistically significant. All tests were two-tailed.

## Figures and Tables

**Figure 1 ijms-22-06147-f001:**
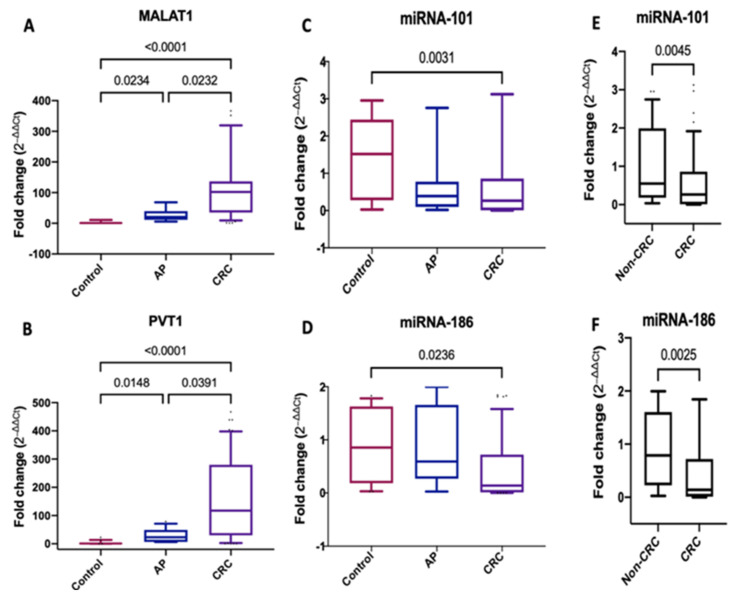
Serum expression levels of *MALAT1*, *PVT1*, *miRNA-101* and *miRNA-186*. (**A**–**D**) Fold change of serum *MALAT1*, *PVT1*, *miRNA-101* and *miRNA-186* expression levels in patients with CRC *(n* = 140) and adenomatous polyps (*n* = 40) compared with healthy controls (*n* = 100). (**E**,**F**) Fold change of *miRNA-101* and *miRNA-186* expression levels in CRC (*n* = 140) versus non-CRC (*n* = 140). For the control samples, the 2^−∆∆ct^ was calculated by subtracting each control value from the average control. Data were expressed as box blot; the box represents the 25–75% percentiles; the line inside the box represents the median and the upper and lower lines representing the 10–90% percentiles. *p* < 0.05 means statistical significance.

**Figure 2 ijms-22-06147-f002:**
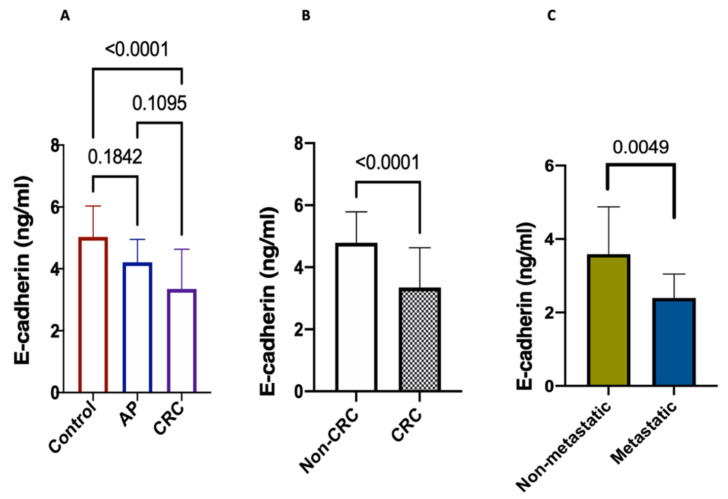
Serum levels of E-cadherin. Data expressed as mean ± SD. *p* < 0.05 means statistical significance. (**A**) CRC (*n* = 140), adenomatous polyps (*n* = 40) and healthy controls (*n* = 100), (**B**) CRC (*n* = 140) vs. non-CRC groups (*n* = 140), (**C**) metastatic (*n* = 24) vs. non-metastatic (*n* = 116).

**Figure 3 ijms-22-06147-f003:**
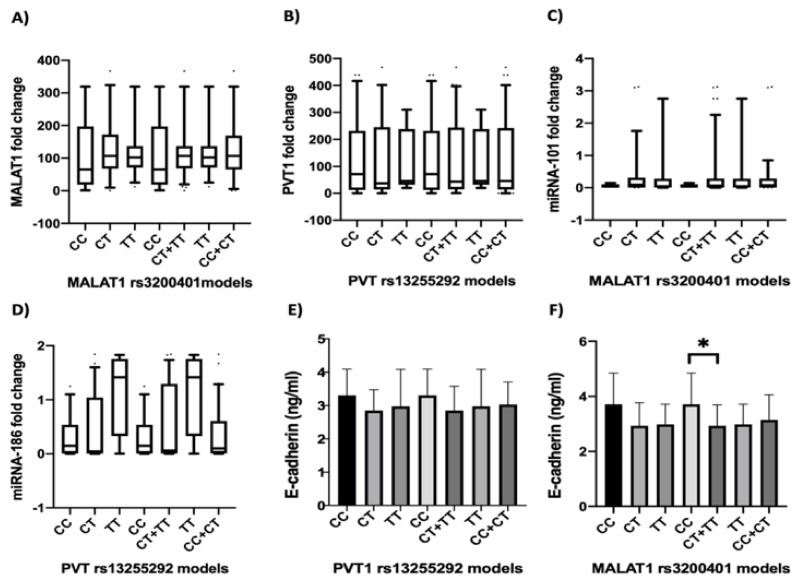
Serum *MALAT1*, *PVT1*, *miRNA-101*, *miRNA-186* and E-cadherin expression levels in CRC patients with different *rs3200401* and *rs13255292* genotypes. The box represents the 25%–75% percentiles; the line inside the box represents the median and the upper and lower lines representing the 10%–90% percentiles of the fold change of serum (**A**) *MALAT1I*, (**B**) *PVT1*, (**C**) *miRNA-101* and (**D**) *miRNA-186*. (**E**,**F**) The bars represent the mean and SD of serum E-cadherin.* means statistical significance *p* < 0.05.

**Figure 4 ijms-22-06147-f004:**
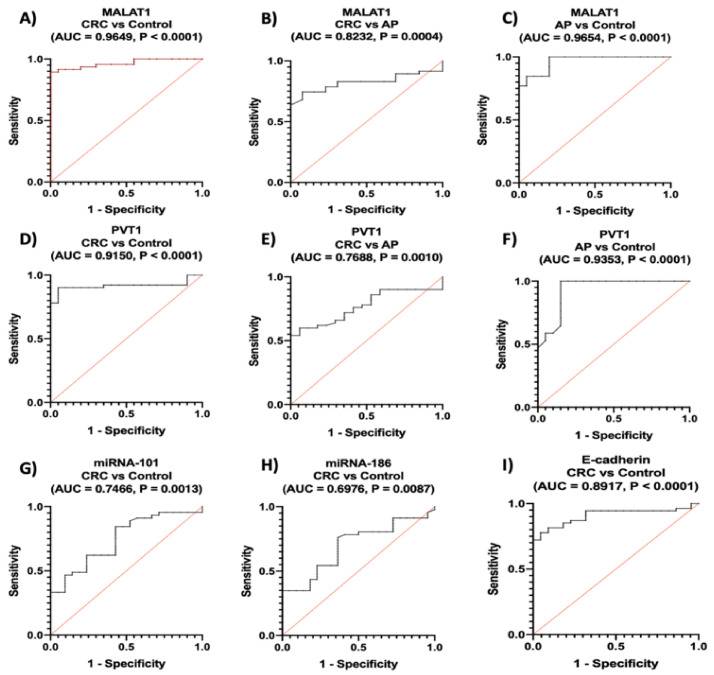
Diagnostic performance of serum (**A**–**C**) *MALAT1*, (**D**–**F**) *PVT1*, (**G**) *miRNA-101*, (**H**) *miRNA-186* and (**I**) E-cadherin. Using ROC curve analysis, CRC (*n* = 140), AP (*n* = 40), healthy controls (*n* = 100).

**Figure 5 ijms-22-06147-f005:**
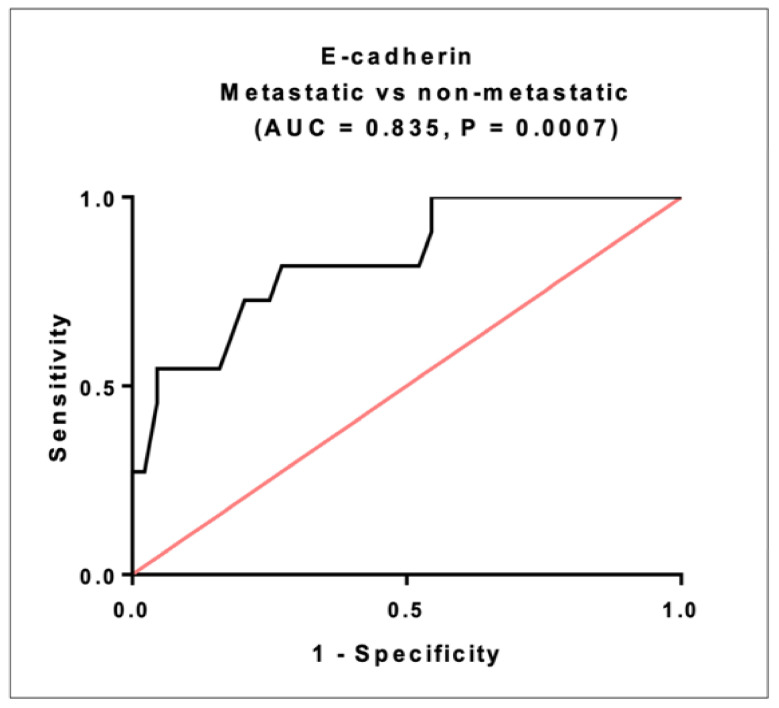
The prognostic performance of serum E-cadherin. Using ROC curve analysis, metastatic patients (*n* = 24), non-metastatic patients (*n* = 116).

**Figure 6 ijms-22-06147-f006:**
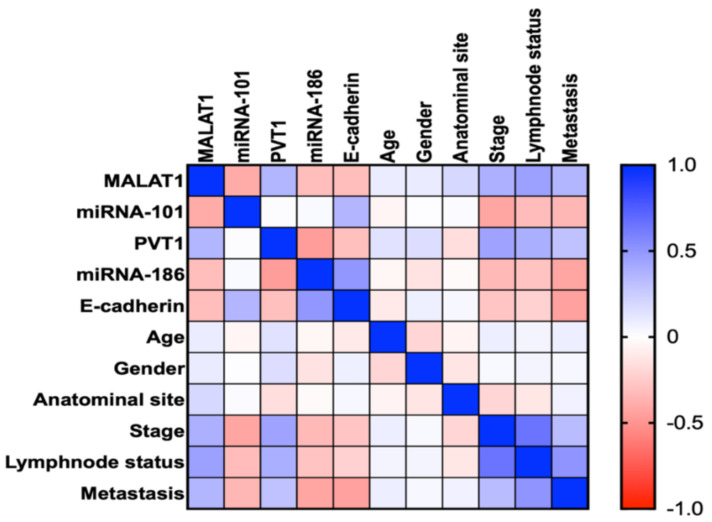
Correlations between studied serum markers with each other and with clinical data in CRC group. A correlation map with a blue-red (cold-hot) scale. The blue color corresponds to a correlation close to 1 and the red color corresponds to a correlation close to −1. White corresponds to a correlation close to 0. Correlations are made by spearman correlation.

**Table 1 ijms-22-06147-t001:** Values of the demographic and clinicopathological data of the studied groups.

		CRC (*n* = 140)	AP (*n* = 40)	Healthy Controls (*n* = 100)	*p* Value
Age (years) Age range		50 ± 12 (25–73)	37 ± 16 (19–75)	49 ± 8.2 (30–65)	**<0.0001 ^a^** **0.888 ^b^** **0.006 ^c^**
**Sex, *n* (%)**	Male	90 (64.3%)	24 (60%)	62 (62%)	0.861
	Female	50 (35.7%)	16 (40%)	38 (38%)	
Smokers		42 (30%)	8 (20%)	19 (19%)	0.084
Hemoglobin (g/dL)		10.94 ± 2.75	11.98 ± 1.37	12.22 ± 1.45	**0.04 ^a^** **0.009 ^b^** **0.88 ^c^**
Platelet count × 10^3^/mm^3^		276.6 ± 94.48	261.1 ± 40	248.6 ± 39.98	0.156
TLC × 10^3^/mm^3^		6.98 ± 2.63	5.66 ± 1.05	6.44 ± 1.33	0.071
ESR (mm/h)		46 ± 31.38	22 ± 15	20.1 ± 11	**<0.001 ^a,b^** **>0.99**
**Anatomical Site, *n* (%)**					
Proximal colon		58 (41.4%)	8 (20%)	-	-
Distal colon		45 (32.1%)	17 (42.5%)	-	-
Rectum		37 (26.5%)	15 (37.5%)	-	-
**Stage, *n* (%)**					
Stage I, II (Early)		88 (62.8%)	-	-	-
Stage III, IV (Late)		52 (37.2%)	-	-	-
**Tumor Grade, *n* (%)**					
Well-differentiated		26 (18.6%)	-	-	-
Moderately differentiated		98 (70%)	-	-	-
Poorly-differentiated		16 (11.4%)	-	-	-
**Lymphatic Metastasis, *n* (%)**					
Present		52 (37.1%)	-	-	-
Absent		85 (57.8%)	-	-	-
Cannot be Assessed		3 (2.1%)	-	-	-
**Distant Metastasis, *n* (%)**					
Present		24 (17.1%)	-	-	-
Absent		116 (82.9%)	-	-	-
**Tissue Type**					
Adenocarcinoma		131 (93.5%)	-	-	-
Non-adenocarcinoma		9 (6.5%)	-	-	-

Data of the studied groups are expressed as mean ± SD or number (percentage). AP: adenomatous polyps; CRC: colorectal cancer; ESR: erythrocyte sedimentation rate; TLC: total leukocyte count. *p* values in bold are statistically significant (*p* < 0.05). ^a^ CRC vs. AP. ^b^ CRC vs. Control. ^c^ AP vs. Control.

**Table 2 ijms-22-06147-t002:** Genotype and Allele Frequencies of *MALAT1 rs3200401* (*C/T*) and *PVT1 rs13255292* (*C/T*) in CRC and Healthy Controls.

*MALAT1 rs3200401* (*C*/*T*) (*n* = 280, Adjusted Analysis)							
Model	Genotype	Control (*n* = 100)	CRC (*n* = 140)	Adjusted OR (95% CI)	*p* Value ^a^	AIC ^a^	BIC ^a^
**Codominant**	*CC*	42 (42%)	31 (22.1%)	1	<0.0001	306	323.4
	*CT*	48 (48%)	62 (44.3%)	1.78 (0.97–3.27)			
	*TT*	10 (10%)	47 (33.6%)	**6.79 (2.92–15.80)**			
**Dominant**	*CC*	42 (42%)	31 (22.1%)	1	<0.0001	316.3	330.2
	*CT-TT*	58 (58%)	109 (77.9%)	**2.62 (1.48–4.64)**			
**Recessive**	*CC-CT*	90 (90%)	93 (66.4%)	1	<0.0001	307.5	321.4
	*TT*	10 (10%)	47 (33.6%)	**4.82 (2.25–10.31)**			
**Overdominant**	*CC-TT*	52 (52%)	78 (55.7%)	1	0.6	327	341
	*CT*	48 (58%)	62 (44.3%)	0.87 (0.51–1.47)			
**Allelic**	*C*	132 (53%)	124 (44%)	**2.43 (1.64–3.61)**	<0.0001	305.8	319.7
	*T*	68 (34%)	156 (56%)				
***PVT1 rs13255292* (*C*/*T*) (*n* = 280, Adjusted Analysis)**							
**Model**	**Genotype**	**Control (*n* = 100)**	**CRC (*n* = 140)**	**Adjusted OR (95% CI)**	***p* Value ^a^**	**AIC ^a^**	**BIC ^a^**
**Codominant**	*CC*	31 (31%)	64 (45.7%)	1	0.086	324.4	341.8
	*CT*	55 (55%)	62 (44.3%)	**0.56 (0.31–0.98)**			
	*TT*	14 (14%)	14 (10%)	0.50 (0.21–1.19)			
**Dominant**	*CC*	31 (31%)	64 (45.7%)	1	0.028	322.5	336.4
	*CT-TT*	69 (69%)	76 (54.3%)	**0.54 (0.32–0.94)**			
**Recessive**	*CC-CT*	86 (86%)	126 (90%)	1	0.38	326.6	340.5
	*TT*	14 (14%)	14 (10%)	0.70 (0.31–1.56)			
**Overdominant**	*CC-TT*	45 (45%)	78 (55.7%)	1	0.12	324.8	338.8
	*CT*	55 (55%)	62 (44.3%)	0.66 (0.39–1.11)			
**Allelic**	*C*	117 (58%)	190 (68%)	**0.66 (0.44–0.98)**	0.04	323.1	337
	*T*	83 (42%)	90 (32%)				

Values are expressed as number (percentage). ^a^ adjusted for age and sex in a logistic regression model. *p* < 0.05 means statistical significance. AIC: Akaike’s Information Criteria, BIC: Bayesian Information Criteria.

**Table 3 ijms-22-06147-t003:** Genotype and Allele Frequencies of *MALAT1 rs3200401* (*C*/*T*) and *PVT1 rs13255292* (*C*/*T*) in CRC and Non-CRC.

*MALAT1 rs3200401* (*C*/*T*) (*n* = 280, Adjusted Analysis)							
Model	Genotype	Non-CRC (*n* = 140)	CRC (*n* = 140)	Adjusted OR (95% CI)	*p* Value ^a^	AIC ^a^	BIC ^a^
**Codominant**	*CC*	59 (42.1%)	31 (22.1%)	1	<0.0001	362.2	380.4
	*CT*	63 (45%)	62 (44.3%)	**1.82 (1.03–3.23)**			
	*TT*	18 (12.9%)	47 (33.6%)	**5.12 (2.49–10.51)**			
**Dominant**	*CC*	59 (42.1%)	31 (22.1%)	1	<0.0001	369.8	384.4
	*CT-TT*	81 (57.9%)	109 (77.9%)	**2.53 (1.48–4.32)**			
**Recessive**	*CC-CT*	122 (87.1%)	93 (66.4%)	1	<0.0001	364.5	379
	*TT*	18 (12.9%)	47 (33.6%)	**3.60 (1.92–6.77)**			
**Overdominant**	*CC-TT*	77 (55%)	78 (55.7%)	1	0.83	381.8	396.3
	*CT*	63 (45%)	62 (44.3%)	0.95 (0.58–1.54)			
**Allelic**	*C*	181 (65%)	124 (44%)	**2.21 (1.55–3.14)**	<0.0001	360.9	375.4
	*T*	99 (35%)	156 (56%)				
***PVT1 rs13255292* (*C*/*T*) (*n* = 280, Adjusted Analysis)**							
**Model**	**Genotype**	**Non-CRC (*n* = 140)**	**CRC (*n* = 140)**	**OR (95% CI)**	***p* Value**	**AIC**	**BIC**
**Codominant**	*CC*	47 (33.6%)	64 (45.7%)	1	0.1	379.3	397.4
	*CT*	75 (53.6%)	62 (44.3%)	0.59 (0.35–1.00)			
	*TT*	18 (12.9%)	14 (10%)	0.55 (0.24–1.24)			
**Dominant**	*CC*	47 (33.6%)	64 (45.7%)	1	0.034	377.3	391.9
	*CT-TT*	93 (66.4%)	76 (54.3%)	**0.59 (0.36–0.96)**			
**Recessive**	*CC-CT*	122 (87.1%)	126 (90%)	1	0.42	381.2	395.7
	*TT*	18 (12.9%)	14 (10%)	0.73 (0.34–1.57)			
**Overdominant**	*CC-TT*	65 (46.4%)	78 (55.7%)	1	0.12	379.4	393.9
	*CT*	75 (53.6%)	62 (44.3%)	0.68 (0.42–1.10)			
**Allelic**	*C*	169 (60%)	190 (68%)	0.70 (0.49–1.01)	0.048	377.9	392.5
	*T*	111 (40%)	90 (32%)				

Values are expressed as number (percentage). ^a^ adjusted for age and sex in a logistic regression model. *p* < 0.05 means statistical significance. AIC: Akaike’s Information Criteria, BIC: Bayesian Information Criteria.

**Table 4 ijms-22-06147-t004:** Association of haplotypes with CRC risk.

Haplotype		Total Frequency	Frequency in Non-CRC Group (*n* = 140)	Frequency in CRC Patients (*n* = 140)	Adjusted OR (95% CI)	*p* Value
*PVT1 rs13255292*	*MALAT1 rs3200401*					
*C ^a^*	*C*	0.3574	0.404	0.3185	1.00	-
*C ^a^*	*T ^a^*	0.2836	0.1995	0.3601	**2.21 (1.31–3.72)**	0.0032 *
*T*	*C*	0.1872	0.2424	0.1244	0.64 (0.34–1.20)	0.17
*T*	*T ^a^*	0.1717	0.154	0.197	1.50 (0.89–2.51)	0.13

Adjusted by age and sex in a logistic regression model using SNPstats online software. Global haplotype association *p* value < 0.0001. ^a^ risk allele. * Statistically significant *p* < 0.05.

**Table 5 ijms-22-06147-t005:** Diagnostic accuracy of the studied markers between CRC and non-CRC groups.

Marker	Cutoff	AUC	*p* Value	Sensitivity	Specificity	PPV	NPV	95% CI
***MALAT1***	>23.71-fold	**0.907**	<0.0001	**82%**	**88%**	90%	78%	0.8435 to 0.9707
***PVT1***	>13.96-fold	**0.848**	<0.0001	**90%**	**70%**	80%	84%	0.7641 to 0.9316
***miRNA-101***	<0.28-fold	0.686	0.0049	62%	67%	72%	56%	0.5697 to 0.8015
***miRNA-186***	<0.20-fold	0.702	0.0028	54%	80%	78%	55%	0.5861 to 0.8172
***E-Cadherin***	<3.81 ng/mL	**0.864**	<0.0001	**76%**	**90%**	91%	73%	0.7837 to 0.9442

AUC—area under the curve, (PPV)—positive predictive value, (NPV)—negative predictive value, (CI)—confidence interval. *p* < 0.05 means statistical significance.

**Table 6 ijms-22-06147-t006:** Logistic regression analysis to predict the risk of CRC in non-CRC groups.

Parameter	Beta Coefficient	SE	*p* Value	OR	OR (95% CI)
Univariate					
***MALAT1***	0.05	0.013	**<0.0001**	1.052	1.052–1.078
***PVT1***	0.0316	0.376	**0.0027**	1.032	1.011–1.053
***miRNA-101***	−0.54	0.254	**0.0325**	0.58	0.3529–0.955
***miRNA-186***	−1.12	0.39	**0.0039**	0.324	0.1512–0.697
***E-Cadherin***	−1.26	0.313	**<0.0001**	0.2811	0.1523–0.518
**Multivariate**					
***MALAT1***	0.0398	0.015	**0.0064**	1.0403	1.0112–1.070
***PVT1***	0.0187	0.011	0.0877	1.0188	0.9973–1.040
***miRNA-101***	−1.3931	0.759	0.0664	0.2483	0.0561–1.386
***miRNA-186***	−1.2410	0.750	0.0980	0.2891	0.0665–1.208
***E-Cadherin***	−0.7858	0.450	0.0805	0.4532	0.1856–1.106
Constant	2.62				

Log likelihood of the stepwise multivariate logistic regression model = −17.966, −2 Log likelihood =35.9339, *p* < 0.0001. *p* value in multivariate analysis adjusted for age and sex. CRC, *n* = 140, non-CRC (Healthy controls + AP), *n* = 140. *p* values in bold are statistically significant *p* < 0.05.

## Data Availability

All data generated or analyzed during this study are included in this published article and its Appendix A.

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
