# Peer review of "Association of MALAT1 and PVT1 Variants, Expression Profiles and Target miRNA-101 and miRNA-186 with Colorectal Cancer: Correlation with Epithelial-Mesenchymal Transition"

_ijms, 2021, doi:10.3390/ijms22116147_

Round 1
Reviewer 1 Report
In this study, Radwan et al. investigated the impact of MALAT1 rs3200401 and PVT1 rs13255292 on CRC susceptibility, their relationship with clinicopathological features, and their correlation with serum MALAT1, PVT1, miRNA-101, miRNA-186 and E-cadherin.
Despite this is a well-written manuscript, I have some considerations:
- the methodological approach and the structure of this work is very similar to a previously published paper written by the same authors focused on other SNPs and lncRNAs (Shaker OG, Senousy MA, Elbaz EM. Association of rs6983267 at 8q24, HULC rs7763881 polymorphisms and serum lncRNAs CCAT2 and HULC with colorectal cancer in Egyptian patients. Sci Rep. 2017;7(1):16246. Published 2017 Nov 24. doi:10.1038/s41598-017-16500-4). I suggest to improve this manuscript avoiding similarities as much as possible.
- The lack of statistical association of rs3200401 and rs13255292 with serum MALAT1, PVT1, miRNA-101, miRNA-186 and E-cadherin, limits the role of these two SNPs as genetic biomarkers of CRC. This should be better highlighted in the discussion.
- It would be interesting to perform also haplotype analysis.
- Additional information on the tested SNPs should be included in the introduction or discussion (i.e. position: intron, exon, promoter region, 5’-UTR, 3’-UTR).
- I suggest to change graph type in Figure 3 A-D and the resolution of all figures.
Author Response
"Please see the attachment."

Reviewer 2 Report
- The current work is an impressive clinical study which interrelates MALAT1 and PVT1 and also miR-101 and miR-186 with CRC.
- miR's chosen in the current study deserve some briefing in the introduction as they have a great potential in therapeutics as targets.
- Also, in the results section Serum levels of E-cadherin", I advise the authors to add a little description about how have they actually differentiated metastatic and non metastatic groups (criteria) for the readers.
- Other significant correlations have been well related by the team signifying the importance of MALATI, PVT1 and miR's.
- There are few minor typographical errors in the manuscript for which the authors are advised to look after,
Author Response
"Please see the attachment."
